# Sol-Gel Derived Tungsten Doped VO_2_ Thin Films on Si Substrate with Tunable Phase Transition Properties

**DOI:** 10.3390/molecules28093778

**Published:** 2023-04-27

**Authors:** Xiaoming Ding, Yanli Li, Yubo Zhang

**Affiliations:** 1AECC Beijing Institute of Aeronautical Materials, Beijing 100095, China; 2National Innovation (Qingdao) High Speed Train Material Research Institute Co., Ltd., Qingdao 370214, China; 3Department of Materials Engineering, Sichuan Engineering Technical College, Deyang 618000, China

**Keywords:** vanadium dioxide, W doping, sol-gel, phase transition

## Abstract

Vanadium dioxide (VO_2_) with semiconductor-metal phase transition characteristics has presented great application potential in various optoelectrical smart devices. However, the preparation of doped VO_2_ film with a lower phase transition threshold on Si substrate needs more investigation for the exploration of silicon-based VO_2_ devices. In this work, the VO_2_ films doped with different contents of W element were fabricated on high-purity Si substrate, assisted with a post-annealing process. The films exhibited good crystallinity and uniform thickness. The X-ray diffraction and X-ray photoelectron spectroscopy characterizations illustrated that W element can be doped into the lattice of VO_2_ and lead to small lattice distortion. In turn, the in situ FT-IR measurements indicated that the phase transition temperature of the VO_2_ films can be decreased continuously with W doping content. Simultaneously, the doping would lead to largely enhanced conductivity in the film, which results in reduced optical transmittance. This work provides significant insights into the design of doped VO_2_ films for silicon-based devices.

## 1. Introduction

Phase transition oxides have attracted great attention due to the rich physics of phase transition phenomena and their huge application potential in various optoelectrical devices [1,2]. Vanadium dioxide (VO_2_) is one of the most intriguing prototypes. It exhibits reversible semiconductor-metal phase transition, accompanied by giant and steep changes in resistivity, optical transmission, reflection, etc. Moreover, this phase transition can be triggered by lots of excitation sources, such as temperature, electric field, laser, and strain [3,4,5]. Thus, VO_2_ has been proposed to be available for thermochromic windows, sensors, memristors, uncooled infrared focal planes for thermal imagers, etc. [6,7,8]. Particularly, recent progress in terahertz (THz) technology indicates that VO_2_ is quite suitable for smart devices including modulators, switching, and filters for THz communications and imaging [9,10].

Silicon-based devices are fundamental for the present semiconductor industry. Thus, the preparation of VO_2_ films on a silicon substrate is significant for its further applications. There are already plenty of reported methods for the preparation of VO_2_ films, such as magnetron sputtering, pulse laser deposition, chemical vapor deposition (CVD), and sol-gel method [11,12,13,14]. Wherein, the sol-gel method presents many advantages, such as being simpler and having faster processing, suitable for large-scale deposition, and easy to carry out the composition design. It has been reported that both inorganic and organic sol-gel methods can be optimized to fabricate the VO_2_ film [14,15,16]. Particularly, Shi et al. developed a method to pre-treat the Si substrate with a hydrophilic solution and obtained enhanced hydrophilicity, and then the bonding of Si substrate with precursor V_2_O_5_ gel can be improved largely. It provided a route to fabricate high-quality VO_2_ films by overcoming the contradiction between the hydrophobicity of substrate and hydrophilicity of inorganic sol-gel [14]. Wu et al. proposed a design of organic sol-gel method to fabricate VO_2_ film, which was available for various hydrophobic substrates [16]. Furthermore, the sol-gel method shows unique convenience in the design and fabrication of VO_2_ films doped with tungsten (W), molybdenum (Mo), titanium (Ti), etc. Particularly, the VO_2_ films doped with W element can decrease the phase transition threshold (temperature, laser pump fluence, etc.) remarkably [17,18,19,20,21], which paves the way for the application of VO_2_ film in low-power optoelectrical devices. Nevertheless, the preparation of W-doped VO_2_ film on Si substrate using the sol-gel method has been reported rarely.

In this work, we used an inorganic sol-gel method to fabricate VO_2_ films doped with different contents of W element. The precursor sol containing W element was designed. Additionally, the Si substrate was pre-treated using a reported hydrophilic treatment process, which then exhibited good compatibility with the sol. The gel films were annealed for the crystallization and stoichiometry evolution of vanadium oxides to form the VO_2_ phase eventually. The films presented tunable phase transition temperature and optical switching properties. The results would be significant for the fabrication and application of silicon-based VO_2_ devices.

## 2. Experimental Section

**Preparation of W-doped VO_2_ films on Si substrate**. The VO_2_ films doped with different contents of W element were fabricated using an inorganic sol-gel method. Firstly, the ammonium tungstate ((NH_4_)_5_H_5_[H_2_(WO_4_)_6_]•H_2_O) and V_2_O_5_ powder were mixed and molted at around 850 °C. Subsequently, the precursor sol was fabricated by pouring the molten mixed powder into deionized (DI) water, with the proportion of V_2_O_5_ 1 g/DI water 40 mL. The single crystal Si (100) substrates (~2000 Ω cm resistivity) were pre-treated with ethyl alcohol and hydrophilic solutions subsequently [22]. Then, the precursor films were spin-coated on Si substrate and annealed at around 500 °C at nitrogen atmosphere for 1.5 h. In this process, the crystallization and phase evolution of vanadium oxides will occur and lead to the formation of W-doped VO_2_ film.

**Characterization**. The crystalline structures of the films were analyzed by X-ray diffraction (XRD, X’ Pert, Philips, Amsterdam, The Netherlands) with Cu Kα (λ = 0.154056 nm) radiation source. The morphologies of products were investigated by scanning electron microscopy (SEM, S-4800, Hitachi, Tokyo, Japan). Additionally, the thickness of the film was determined by the cross-sectional SEM morphology. The vanadium valence states and chemical composition of the VO_2_ thin films were detected by X-ray photoelectron spectroscopy (XPS, Kratos, Manchester, UK) using Al Kα (h*v* = 1486.6 eV) exciting source. The optical properties of the films were investigated by Tensor 27 (Bruker, Bremen, Germany) spectrometer attached with an adapted heating-controlled unit, and then the hysteresis loops of VO_2_ films were received by collecting the transmittance of films at a fixed wavelength (4 μm). The square resistances of the films were measured using a four-point probe system (280SI) with a controllable heating system.

## 3. Results and Discussion

Figure 1a shows the XRD patterns of the VO_2_ films doped with different concentrations of W element. A strong diffraction peak is observed at the angle 2θ = 27.52° for the undoped VO_2_ film and W-doped VO_2_ films, which can be indexed to the (011) plane of VO_2_. It indicates that the incorporation of W does not affect the preferential orientation of VO_2_ in the (011) direction on the single crystal Si substrate. No peaks of any other vanadium oxides (such as V_2_O_5_ and V_2_O_3_) are observed, revealing that the products have high purity. Additionally, there are no peaks related to ammonium tungstate or their derivatives, suggesting that the W atoms are incorporated into the crystal lattice of VO_2_ and forming the substitutional solid solution. Furthermore, the magnified image at around 27.52° indicates that the doping with W element will lead to a redshift of the (011) peak with an increased doping amount of W (as shown in Figure 1b). It can be ascribed to the larger atom size of W compared to V, which results in lattice expansion in VO_2_. It should be noted that the precursor films were annealed at around 500 °C in a nitrogen atmosphere for 1.5 h [14,15,16]. This annealing process has been widely reported for obtaining a high-purity VO_2_ phase. In this work, the W-doped films present still demonstrate high quality traits.

The typical SEM morphology of VO_2_ films deposited on the Si substrates with different W doping contents are presented in Figure 2. It is worth noting that W doping has a great influence on the morphology of VO_2_ films. Figure 2a shows the SEM photograph of the film without W doping. Most of the particles are bonded together, and boundaries can hardly be seen. Moreover, the film shows obvious microcracks which are harmful to the phase transition properties of VO_2_ films [23]. For the sample that had a W-doping of 0.61%, the film is uniform and compact with large grains of about 120 nm, and fuzzy boundaries can be seen. With 1.12% doping, the film is more compact, and the grain sizes are reduced slightly compared with 0.61% W-doped VO_2_ film. In addition, the film exhibits more clearly grain boundaries. Subsequently, as the W doping content increases to 1.62%, the grain size of the films is further reduced. This result suggests that W doping could make the VO_2_ film more compact and reduce the grain size. Furthermore, the cross-sectional shapes of the W-doped film were observed, as shown in Figure 3, displaying the thickness of the film of about 404.3 nm.

XPS was performed to investigate the composition and chemical state of W-doped VO_2_ film deposited on the Si substrates. Figure 4a shows the wide-range survey spectrum of the 1.62% W-doped VO_2_ film. It reveals that the sample consists of vanadium, oxygen, nitrogen, carbon, silicon, and tungsten, where the peak of silicon signals comes from the substrates, and nitrogen and carbon are attributed to the adventurous hydrocarbon contamination on the sample surface. In Figure 4b, the V2p peaks were fitted with a Shirley function. The V2p_3/2_ peak is separated into two peaks, meaning two valence states of vanadium (+4 valence and +5 valence) exist in the sample, where the binding energy of 515.81 eV and 516.98 eV correspond to V^4+^ and V^5+^, respectively. Both of the binding energy for the two valence states of vanadium are consistent with the value in VO_2_ in previous reports [24]. The nonstoichiometry of vanadium in the film can be attributed to oxidation at the surface of the sample when exposed to air. The fractional percentage of the +4 valence state in the VO_2_ film on the Si substrates can be evaluated according to the peak area of V^4+^ and V^5+^, as about 66.3%, illustrating that the main component of W-doped VO_2_ film is VO_2_. Figure 4c shows the binding energies of 35.18 eV and 36.98 eV for W4f_7/2_ and W4f_5/2_, respectively, revealing that the existing form of W ion in this sample is W^6+^ [25,26].

The optical properties of VO_2_ films were investigated by infrared spectra. Figure 5 shows the infrared transmittance at room temperatures for VO_2_ films with W doping concentrations of 0% and 1.62%. Their infrared transmittances at the wavelength of 4 μm are 58% and 40%, respectively. The reason for the transmittance decreases significantly with increasing W doping contents could be as follows: the concentration of carriers generated by the thermal excitation in the band gap of semiconductor VO_2_ increases with the increase of the W doping content, and the infrared shielding effect caused by carrier reduces the infrared transmittance of the VO_2_ films [27].

Figure 6 displays the hysteresis loop of transmittance–temperature at a fixed wavelength of 4 μm for VO_2_ films synthesized with different W-doped contents. The corresponding first-order derivative curves are shown in Figure 6 inset. The figures clearly illustrate the influence of W doping on the phase transition of VO_2_ films. The calculated results from the figures are shown in Table 1. It can be seen from Figure 6 and Table 1, the *T_c_* are 67.35, 47.9, 40.4, and 33.35 °C and hysteresis widths are 9.9, 5.2, 6.0, and 9.3 °C for 0%, 0.61%, 1.12% and 1.62% W-doped VO_2_ films, respectively. Several characteristics can be concluded. First, compare with the undoped film, the *T_c_* of VO_2_ films decrease effectively. With the introduction of ammonium tungstate, the V^4+^-V^4+^ pairs are destroyed by the partial substitution of V atoms with W atoms, which reduces the stability of the structure of VO_2_. Therefore, the phase transition temperature of VO_2_ films reduces [28]. Second, the hysteresis widths increase with the increase of the W doping content. However, all of them are less than the undoped film. In the case of Lopez et al., the driving force of phase transition comes from various defects in the film [29]. According to the research of Jing Du et al., the introduction of W could probably enhance the nucleation density (ρ) of defects, the bulk free energy (Δg_ex_) per unit volume decreases, while Δg_ex_ = c|T−T^0^| (where c is a constant, T^0^ is the *T_c_* for undoped film, T represents the actual *T_c_* of films), |T−T^0^| decreases with decreasing of g_ex_, and smaller value of |T−T^0^| can trigger phase transition, therefore resulting in decreased hysteresis widths [30].

The electrical performance of VO_2_ films was tested using the conventional four-point-probe method. Figure 7 shows the square resistance versus temperature curves of VO_2_ films. It can be seen that the square resistances of the VO_2_ thin films decrease exponentially as the temperature increases. The undoped VO_2_ film has a quite high square resistance of 144 KΩ/□ at 30 °C and 0.28 KΩ/□ at 85 °C. The orders of magnitude of the square resistance transition is approximately 3. For 1.62% W-doped VO_2_ thin film, its square resistance is 64 KΩ/□ at 30 °C and 1.25 KΩ/□ at 70 °C. These results imply that W doping can reduces not only the square resistance of VO_2_ thin films but also the orders of magnitude of square resistance transition, which is consistent with the previous reports [31,32,33]. The 3d^1^ configuration of vanadium ions and the 3d^1^ tied around V^4+^-V^4+^ pairs offer a high activation energy between the conduction and valence band for undoped VO_2_ thin films, resulting in poor conductivity [34]. By doping with W, the partial substitution of V atoms with W atoms favor the enhancement of the electron concentration, and the Fermi level shifts toward the conduction band. Consequently, the activation energy of W-doped VO_2_ thin films decreases, and the conductivity increase [35]. It also can be concluded that the transition temperature of VO_2_ thin films can be tuned by doping W effectively. This is in good agreement with what we have previously obtained from FTIR.

## 4. Conclusions

We used an inorganic sol-gel method to assist with the post-annealing process to fabricate VO_2_ films on a Si substrate. Additionally, the films were doped with different contents of W elements. The results indicated that the films exhibited good crystallinity, uniform thickness, and continuously decreased phase transition temperature with W doping content. However, the doping would lead to enhanced conductivity in the film, which results in reduced optical transmittance. Thus, the balance between lower phase transition temperature and optical switching properties should be considered with regard to the possible devices. This work provides significant insights into the design of doped VO_2_ films for silicon-based devices.

## Figures and Tables

**Figure 1 molecules-28-03778-f001:**
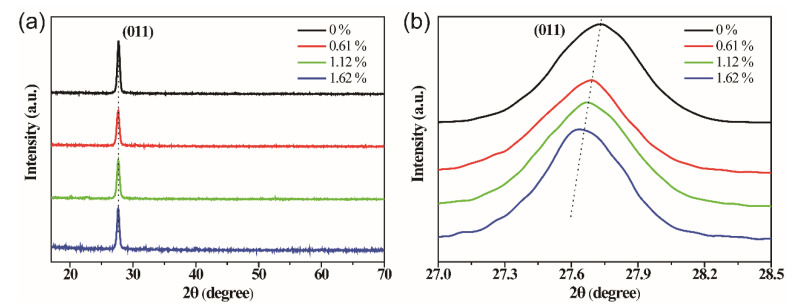
(**a**) XRD patterns of VO_2_ films doped with different concentrations of W element. (**b**) The magnified (011) peak.

**Figure 2 molecules-28-03778-f002:**
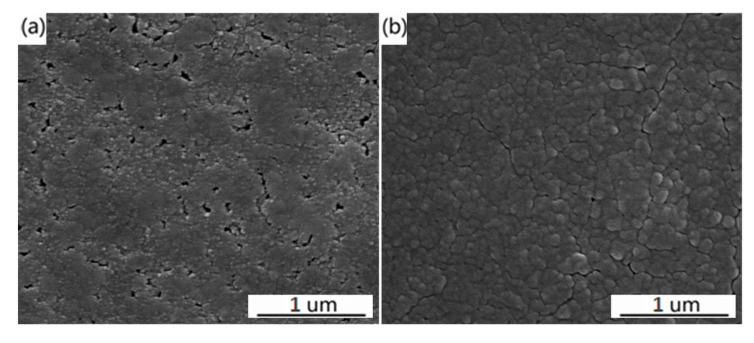
SEM images of the VO_2_ films with different W doping concentrations: (**a**) 0%; (**b**) 0.61%; (**c**) 1.12%; (**d**) 1.62%.

**Figure 3 molecules-28-03778-f003:**
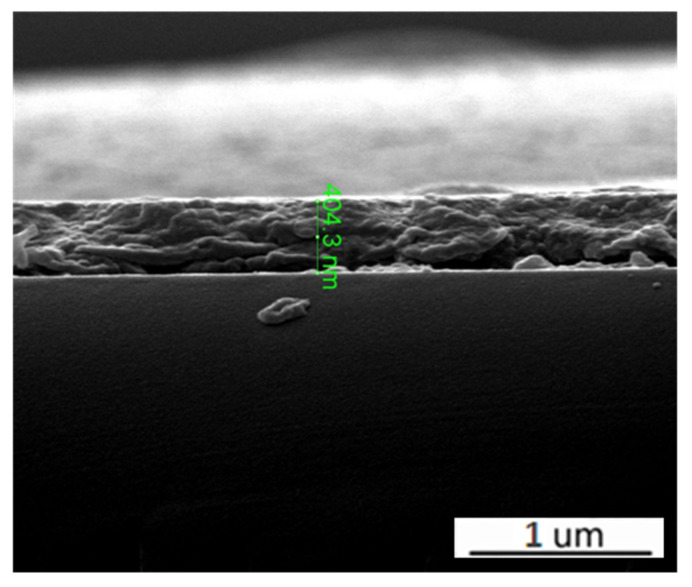
Cross-section shapes of W-doped VO_2_ films.

**Figure 4 molecules-28-03778-f004:**
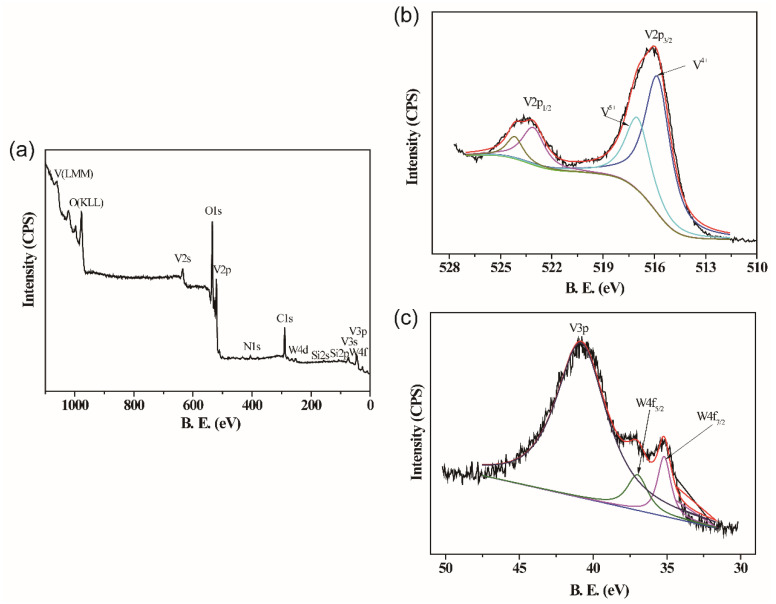
XPS spectra of 1.62% W-doped VO_2_ film: (**a**) survey spectrum, (**b**) core-level spectrum for V2p. The line in black color refers to the initial spectrum, the line in red color refers to the fitting of the spectrum, the line in blue color refers to the fitted peak of V^4+^, and the line in green color refers to the fitted peak of V^5+^, and (**c**) core-level spectrum for W4f. The line in black color refers to the initial spectrum, the line in red color refers to the fitting of the spectrum, the line in purple color refers to the fitted peak of W4f_7/2_, and the line in green color refers to the fitted peak W4f_5/2_.

**Figure 5 molecules-28-03778-f005:**
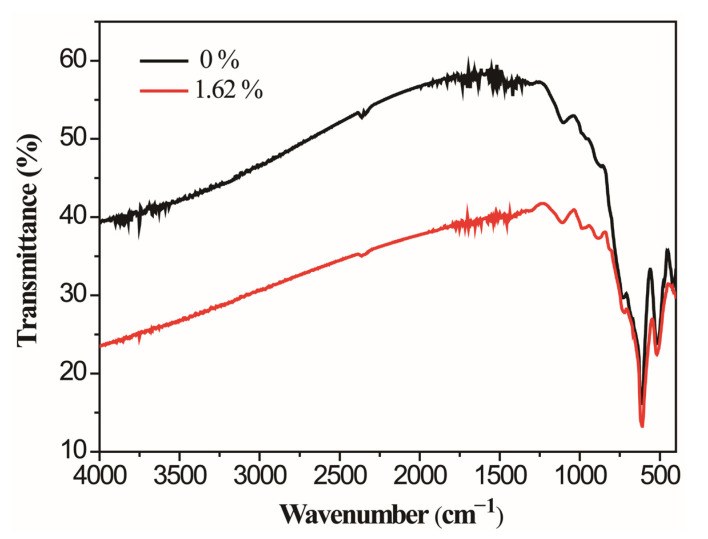
FTIR spectra of the prepared VO_2_ film with different W doping concentrations.

**Figure 6 molecules-28-03778-f006:**
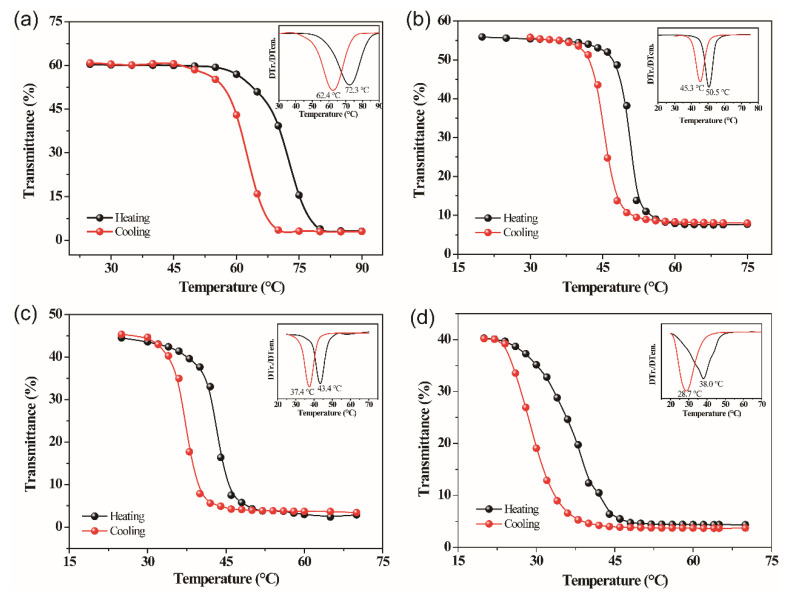
Transmittance hysteresis curves at wavelength of 4 μm and corresponding first order derivative curves for VO_2_ films with different W doping concentration: (**a**) 0%; (**b**) 0.61%; (**c**) 1.12%; (**d**) 1.62%.

**Figure 7 molecules-28-03778-f007:**
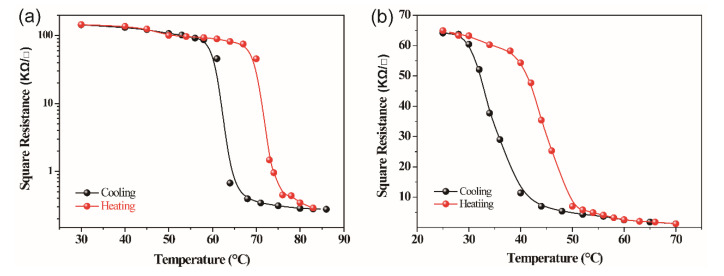
Hysteresis curves of square resistance for undoped VO_2_ films (**a**) and 1.62% W-doped VO_2_ films (**b**).

**Table 1 molecules-28-03778-t001:** Phase transformation properties of W-doped VO_2_ films.

Doping Level (at.%)	*T*_heating_ (°C)	*T*_cooling_ (°C)	*T*_c_ (°C)	Δ*T* (°C)
	Heating	Cooling		
0	72.3	62.4	67.35	9.9
0.61	45.3	50.5	47.9	5.2
1.12	37.4	43.4	40.4	6.0
1.62	28.7	38.0	33.35	9.3

## Data Availability

The data can be found by contacting the corresponding author.

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
