# Peer review of "Sol-Gel Derived Tungsten Doped VO2 Thin Films on Si Substrate with Tunable Phase Transition Properties"

_molecules, 2023, doi:10.3390/molecules28093778_

Round 1

Reviewer 1 Report

This manuscript described synthesis, characterization and showed transmittance hysteresis. Although they use the term of phase transition, I cannot follow the transition from which phase to which phase. 

There are lots of similar articles such as (doi.org/10.1016/S0040-6090(98)00362-9),  (doi.org/10.1016/j.solmat.2007.06.016), and references in this manuscript.

Especially, the results of this work are similar to those of the article (doi.org/10.1016/S0040-6090(98)00362-9).

But, author did not cite this article. I recommend the author to compare lots of previous reports and simply to emphasize the new finding of this work. 

The unit of square resistance is garbled.

corresponding information, keywords, supplmentary materials, funding, sample availability are lacked.

Author Response

Thanks very much for the reviewer's comments. We have revised the manuscript in accordance with the concerns of the reviewer and made careful correction on the technical issues. All changes made to the text are in red color. Here below is our description on revisions according to the comments.

1) This manuscript described synthesis, characterization and showed transmittance hysteresis. Although they use the term of phase transition, I cannot follow the transition from which phase to which phase. 

The author’s answer: Thanks for the reviewer’s comments. The phase transition of VO2 has been widely reported, and it has been recognized that the phase transition triggered by temperature is from semiconductor phase to metal phase. We didn’t carry out the XRD characterization for the film at different temperatures, because it’s already a common phenomenon. Moreover, our results in Figure 6 and 7 clearly indicate the phase transition process, which lead to the changes in infrared transmittance and resistance.

2) There are lots of similar articles such as (doi.org/10.1016/S0040-6090(98)00362-9),  (doi.org/10.1016/j.solmat.2007.06.016), and references in this manuscript. Especially, the results of this work are similar to those of the article (doi.org/10.1016/S0040-6090(98)00362-9). But, author did not cite this article. I recommend the author to compare lots of previous reports and simply to emphasize the new finding of this work. 

The author’s answer: Thanks. The reviewer provided two references, but they are quite different from the work reported in this manuscript. In the first reference (doi.org/10.1016/S0040-6090(98)00362-9), the VO2 films were fabricated by magnetron sputtering, and the doping of W element was realized by ion implantation. In the second reference (doi.org/10.1016/j.solmat.2007.06.016), the W-doped VO2 was not film but particles. By contrast, our work presented the fabrication of W-doped VO2 film using a sol-gel method. We would like to cite these two references to fulfill the Introduction section more.

3) The unit of square resistance is garbled.

The author’s answer: Thanks. We have corrected the unit of square resistance in Figure 7.

4) Corresponding information, keywords, supplmentary materials, funding, sample availability are lacked.

The author’s answer: Thanks for the reviewer’s comment. We have added the request information. But there are not supplementary materials and funding information for this work.

Reviewer 2 Report

The article seems to be interesting due to the development of optoelectronics,
especially in terms of its application.In this work, VO2 films doped with different content of element W were left on a high purity Si foundation to aid annealing.
The conducted research and discussion are useful in the further development
of work on the design of doped VO2 films for silicon devices.
The authors clearly and properly present the results in the form of drawings and analysis. Good quality drawings included. Literature references are correct.

Author Response

Thanks for the reviewer's comments.

Round 2

Reviewer 1 Report

I’m satisfied.